# Pregnant women's attitudes and intentions toward tuberculosis, malaria, group B streptococcus, and respiratory syncytial virus vaccines in pregnant: Findings from a cross-sectional study of pregnant women living in Brazil, Ghana, Kenya, and Pakistan

Rupali Limaye [1,2*], Jessica Schue [1], Berhaun Fesshaye [1], Prachi Singh[1], Emily Miller[1], Renato Souza [3], Saleem Jessani[4], Marleen Temmerman [5], Caroline Dinam Badzi[6], Molly Sauer [1], Vanessa Brizuela [7], Ruth Karron[1]

**1** Department of International Health, Johns Hopkins Bloomberg School of Public Health, Baltimore, Maryland, United States of America, **2** Department of Global and Community Health, George Mason College of Public Health, Fairfax, Virginia, United States of America, **3** Department of Obstetrics and Gynecology, University of Campinas, Campinas, Sao Paulo, Brazil, **4** Department of Community Health Sciences, Aga Khan University, Karachi, Pakistan, **5** Aga Khan University Centre of Excellence in Women and Child Health, Nairobi, Kenya, **6** University of Ghana School of Nursing and Midwifery, Legon, Accra, Ghana, **7** UNDP/UNFPA/UNICEF/WHO/World Bank Special Programme of Research, Development and Research Training in Human Reproduction, Department of Sexual and Reproductive Health and Research, World Health Organization, Geneva, Switzerland

\* rlimaye@jhu.edu

## Abstract

There are numerous infections that can adversely impact a developing fetus, neonates, and pregnant women, and there is limited research related to how specific infections experienced during pregnancy can affect these populations. Tuberculosis (TB), malaria, Group B streptococcus (GBS) and respiratory syncytial virus (RSV) can cause negative outcomes to maternal and neonatal health. For TB and GBS, there are vaccines in various stages of clinical trial development, and malaria and RSV vaccines are available. This study aimed to examine pregnant women's attitudes toward TB, malaria, GBS, and RSV vaccines in Brazil, Ghana, Kenya, and Pakistan. We administered a cross-sectional survey to pregnant women, recruiting women seeking care in primarily urban health facilities. We surveyed 1,603 pregnant women. Participants indicated that vaccine safety for the baby was the most important factor in their decision-making related to vaccine acceptance, followed by vaccine efficacy for the baby, and then vaccine safety for the mother. When asked why they would receive any of the four vaccines, participants indicated that protecting the baby was most important, followed by protecting self, and then stopping the spread of disease. Almost one-third of participants (30%) indicated that they would definitely intend to receive a GBS vaccine, followed by malaria (26%), RSV (25%), and TB

**Data availability statement:** There are ethical and legal restrictions which prevent the public sharing of minimal data for this study, because the data contain potentially identifiable participant information. Data for this study are available upon request via email from IRB contacts at the University of Campinas (cep@unicamp.br), Ghana Health Service (ethics.research@ghs.gov.gh), KEMRI (director@kemri.org), NBC Pakistan (nbcpakistan@nih.org.pk), and JHU IRB (BSPH.irboffice@jhu.edu) for researchers who meet the criteria for access to confidential data.

**Funding:** This study was funded by the Gates Foundation (grant INV-016977 to R.J.L/R.A.K) at the Johns Hopkins Bloomberg School of Public Health. The funders had no role in study design, data collection and analysis, decision to publish, or preparation of the manuscript.

**Competing interests:** The authors have declared that no competing interests exist.

(19%). Related to vaccine hesitancy, approximately 40% of our participants agreed that vaccines are unnatural, 38% agreed that the body should develop natural immunity, and 19% had delayed a recommended vaccine. Pregnant women are interested in receiving various vaccines while pregnant. As several new adult vaccines are on the horizon, understanding the attitudes of potential vaccine beneficiaries at higher risk for diseases is critical for informing clinical trial design and, in the long term, vaccine acceptance.

## Introduction

Many infections pose unique risks to the developing fetus, neonate, and pregnant women [1]. Pregnant women face uncertainty not only during new infectious disease outbreaks, but also in relation to new vaccines in pregnancy. In addition, emerging infectious disease threats during pregnancy are often underrepresented in research, resulting in limited data and suboptimal preparedness planning [2].

One approach to reducing morbidity and mortality among these populations is maternal immunization. Vaccination during pregnancy offers not only protection to the pregnant women against the disease, but also passive immunity to an infant [3,4]. This is through an active process where maternal antibodies are transferred to a fetus through the placenta, offering protection to the infant as soon as it is born [3,4]. Further antibody transfer occurs during breastfeeding, providing additional protection to the infant when they are most vulnerable to infectious diseases [4]. There are several vaccines that are in various stages of clinical trial development, targeted to pathogens that cause substantial negative outcomes related to maternal and neonatal health and are thus priorities for use in pregnant women. These include maternal vaccines in development to target and Group B streptococcus (GBS) and malaria, recently approved maternal respiratory syncytial virus (RSV) vaccines, and future vaccines targeting tuberculosis (TB) [3]. These four diseases are of particular concern in low- and middle-income countries, where disease burden is disproportionately high and alternative prevention or treatment options are less commonly available or accessible.

TB is a significant cause of maternal mortality and morbidity globally [5,6]. Pregnant and postpartum women's increased susceptibility to active TB and accelerated progression of disease may be due to the immunological changes that occur during pregnancy [7,8]. Active TB during pregnancy has also been linked to adverse outcomes for the baby including perinatal death, preterm birth, low birthweight, and fetal distress [6]. While epidemiologic data on TB during pregnancy are lacking due to inconsistent screening of pregnant women, a global burden study estimated that more than 200,000 active TB cases occurred during pregnancy in a single year [5,9]. Currently, the BCG vaccine is the only vaccine for TB prevention and offers protection for infants and young children, but this protection wanes in adolescence; BCG is not recommended in pregnancy [10,11]. Although multiple adult TB vaccines are in clinical development now, these vaccine trials do not currently include pregnant women as trial participants [11].

Malaria presents a significant global health challenge for pregnant women as well as their fetuses and newborns [12]. As with tuberculosis, pregnant women infected with malaria face increased morbidity and mortality compared to non-pregnant women [12,13]. Pregnant women infected with malaria may suffer from anemia, which in turn increases the risk of maternal mortality, whereas risks in-utero and to the infant include miscarriage, premature delivery, stillbirth, low birthweight, and cognitive-developmental delays [12,14–16]. The World Health Organization approximates that in 2022, in countries where malaria transmission is considered to be moderate to high, more than one-third (36%) of pregnancies were exposed to malaria [17]. Each year, 125 million pregnant women are at risk of malarial infection [18]. In December 2024, the Sanaria PfSPZ malaria vaccine was the world's first vaccine to be shown effective for malaria prevention before and during pregnancy [19]. An additional candidate product in development to prevent malaria during pregnancy is a VAR2CSA-based vaccine [20]. As of the writing of this paper, there are currently no approved vaccines for malaria prevention during pregnancy.

GBS is a bacterium that colonizes the vagina and rectum of pregnant people and is usually asymptomatic, but can occasionally cause severe disease in both the pregnant person and the neonate via passage in utero or during vaginal delivery [21]. GBS disease can cause maternal infection and sepsis, stillbirth, preterm birth, and early- and late-onset sepsis and meningitis in infants, which can be fatal [21–23]. In 2020, GBS was estimated to have colonized approximately 20 million pregnant people, and caused approximately 400,000 cases of infant disease, 40,000 cases of maternal disease, and 46,000 stillbirths [24]. Intrapartum antibiotic prophylaxis (IAP) is currently the only available GBS disease prevention strategy, but it is programmatically complex to implement, only effective against early-onset GBS disease in infants, raises concerns about antimicrobial resistance, and is primarily an option in higher income settings only [25,26]. A GBS vaccine for pregnant women is prioritized for its potential to prevent both early and late-onset disease by providing transplacental antibody protection to newborns [27]. Two maternal GBS vaccines by Pfizer and MinervaX are currently in late-stage clinical development [26,27]. Pfizer's GBS6 vaccine is a hexavalent conjugate vaccine targeting the six serotypes that produce 98% of disease [26]. MinervaX's protein-only vaccine, GBS-NN/NN2, is intended to target all clinically significant isolates of GBS [3].

Limited data exist on RSV infection in pregnancy. A 2022 meta-analysis of five studies found a pooled prevalence for RSV of 0.2 per 100 pregnancies, but estimates varied widely across study countries [28]. RSV infection in pregnancy increases the risk of severe maternal disease, especially in those with preexisting lung conditions or coinfections [29]. RSV infection during pregnancy may also be associated with an increased risk of preterm delivery and/or low birth weight in the infant [28]. The impact of RSV on infants is well documented in global health literature; RSV is one of the primary causes of lower respiratory tract infection in children under the age of five, contributing to approximately 33 million infections and 101,400 deaths globally in 2019, with 95% of infections and 97% of deaths occurring in low- and middle-income countries [30]. Infants are at highest risk for severe RSV disease if they enter their first RSV season within the first six months of life [31]. Currently, two products are available to prevent severe RSV disease in infants: RSV monoclonal antibodies administered directly to infants, and an RSV prefusion F maternal vaccine administered to pregnant people that provides immunity to the infant for the first six months of life via placental antibody transfer [32]. As of late 2024, either or both products have been approved in more than 50 primarily high- and middle-income countries, but not in any low-income or lower-middle-income countries outside of India, despite their much greater RSV disease burden [32].

TB, malaria, GBS, and RSV cause negative maternal and neonatal health outcomes, and there are vaccines either available or in development for each of these diseases, including some specifically intended for use in pregnancy. As vaccination, not vaccines, save lives, the objective of this cross-sectional study was to understand and compare attitudes toward these potential vaccines among pregnant women receiving antenatal care in Brazil, Ghana, Kenya, and Pakistan. Individuals weigh multiple factors when deciding whether to be vaccinated and these attitudes and intentions are complex and both vaccine- and context-specific; vaccination in pregnancy presents even more complicated processes as pregnant

women consider the benefits and risks to themselves and their babies [33,34]. The results of this study are intended to inform demand generation efforts for vaccine acceptance, understanding that such efforts should be context specific.

## Methods

### Ethics statement

This study was approved by the following institutional review boards: Johns Hopkins Bloomberg School of Public Health Institutional Review Board (00020864), University of Campinas/Unicamp Brazil (63968222.1.1001.5404), Jundiai University Institutional Review Board Brazil ((63968222.1.1001.5404), Ghana Health Service Ethics Review Committee (028/03/23), Aga Khan University Kenya (2023/ISERC-17(v2)), Pumwani Maternity Hospital (PMH/CEO/76/0785/2023), Aga Khan University Pakistan (2024-8633-30122), National Bioethics Committee for Research Pakistan (4–87/NBCR-1029/23/1087), and Ethics Research Committee World Health Organization (CERC.0193A, CERC.0193B, CERC.0193C), Ethics Review Committee Pan American Health Organization (PAHOERC.0633.01). Formal written consent was obtained for each participant.

### Participants, study setting, and recruitment

This was part of a multi-country, cross-sectional, mixed-methods study using qualitative interviews and quantitative surveys to collect data on factors affecting COVID-19 vaccine decision making among pregnant and postpartum women in Brazil, Ghana, Kenya, and Pakistan. The study protocol and instruments are available online [35]. This analysis includes questions about future maternal vaccines from the quantitative surveys and focused on vaccine intentions related to these diseases in pregnancy. Study sites were located across the four countries, with health facilities included in the study serving a variety of different patient populations. In total, data were collected in nine facilities across the four countries. In Brazil, the research was carried out in two maternity hospitals in the urban São Paulo region: CAISM/Unicamp Hospital in Campinas and Hospital Universitario de Jundiaí in Jundiaí. Data were collected between August 21, 2023 and December 4, 2023. In Ghana, three hospitals in the Greater Accra Region were included: Ga West Hospital, Tema General Hospital, and Shai-Osuduko Hospital. Data were collected between October 25,2023 and November 10, 2023. In Kenya, the study was conducted at two antenatal clinics based in referral hospitals of Nairobi: Aga Khan University Hospital and Pumwani Maternity Hospital. Data were collected between October 13, 2023 and March 8, 2024. In Pakistan, two hospitals in Karachi were involved: Aga Khan Hospital for Women and Children, Kharadar and the Jinnah Postgraduate Medical Center OBGYN Department. Data were collected between February 26, 2024 and May 25, 2024.

We aimed to survey 400 pregnant individuals in each country, for a total of 1,600 surveys. The sample size was determined with 80% power to find differences in proportions of respondents with a positive or negative attitude toward vaccines in pregnancy between two groups with a 5% margin of error. We sought to sample approximately equal numbers by pregnancy trimester. Surveys were administered to pregnant individuals seeking maternity care. Recruitment strategy varied by country, with sites recruiting from waiting areas in clinics using consecutive sampling. In Brazil, every *n*th person in the waiting area was approached, with *n* varying based on patient volume per site. Ghana, Kenya, and Pakistan used a consecutive sampling approach to recruit from either waiting or reception areas.

### Data collection

Trained study staff approached women and read a recruitment script to assess eligibility: 1) study interest, 2) age 18 or older (or an emancipated minor - in Brazil only), 3) fluency in the local language or English, and 4) having heard of the COVID-19 vaccine. Eligible participants interested in joining the study went through a written informed consent process and were provided with written participation information sheets. Study staff paused any study activities if the participant was called to see a provider and restarted after the visit was complete. Ghana, Kenya, and Pakistan provided

transportation remuneration or a food box to participants after administration of the survey; Brazil did not provide any remuneration.

The survey instrument was developed to identify attitudinal, behavioral, intentional, and psychosocial correlates of vaccine behavior. The instrument was developed by using a comprehensive and iterative process that included a literature review and review of relevant instruments. The instrument was then reviewed by country teams, pre-tested, and then finalized and aligned with the local contexts. Each country's data collection team was trained separately by members of the team from Johns Hopkins Bloomberg School of Public Health through a three-day training session. Training sessions included information and practice related to human subject research ethics, quantitative survey best practices, and study instrument practice.

Data collection included a single questionnaire, with Brazil and Kenya using paper-based data collection and Ghana and Pakistan using tablet-based data collection using either the REDCap Mobile Application or REDCap's web-based data entry interface [36,37]. All study data were managed and stored using REDCap electronic data capture tools hosted at Johns Hopkins Bloomberg School of Public Health. Data collection was done in Brazilian Portuguese in Brazil; Ga, Twi, or English in Ghana; Kiswahili or English in Kenya; and Urdu in Pakistan. The survey took approximately 30–60 minutes to complete.

**Measures**

The full survey is publicly available [38]. The sections used in this analysis are described below.

**Sociodemographic characteristics (5 items).** We asked participants to indicate their age (18–24, 25–34, 35–49, 50+), marital status (single/never married, married or cohabitating with partner, divorced/widowed/separated), trimester (first trimester 1–12 weeks, second trimester 13–26 weeks, third trimester 27 weeks or more), how many children under the age of 18 they had (none, one, two, three, four or more), and level of education (no formal schooling, less than primary school, primary school completed, secondary/high school completed, college/university completed, post graduate degree completed).

**Decision-making factors related to future maternal vaccines (2 items).** We asked women to identify 3 factors that were most and least important to them related to future maternal vaccines: "*We are interested in understanding what would be most important to you in your decision to receive any new vaccines for pregnant women (for example vaccines against tuberculosis, malaria, Streptococcus B, respiratory syncytial virus) whenever they become available. Which of the following factors are MOST IMPORTANT in informing your decision to take new vaccines. You can choose up to three factors that are most important to you*" (how well the vaccine works in protecting me, how well the vaccine works in protecting my baby, my doctor's recommendation about the vaccine, my family's recommendation about the vaccine, safety of the vaccine for me, safety of the vaccine for my baby). We also asked women to rank in order the 3 least important decision-making factors out of six factors: "*Which of the following factors are LEAST IMPORTANT in informing your decision to take new vaccines. You can choose up to three factors that are least important to you*" (how well the vaccine works in protecting me, how well the vaccine works in protecting my baby, my doctor's recommendation about the vaccine, my family's recommendation about the vaccine, safety of the vaccine for me, safety of the vaccine for my baby).

**Future maternal vaccine intentions (4 items).** We asked about future maternal vaccine intentions related to tuberculosis, malaria, Group B streptococcus, and respiratory syncytial virus using a 3-point scale. For each vaccine, we asked women to rate their intention to receive the vaccine: "*We would like to understand your interest in receiving specific vaccines that are currently being developed and may become available for pregnant people in the future. For each of the following vaccines, if the vaccine were to become available in your country in the future and your doctor recommended it during pregnancy, what would be your intention to receive the vaccine?*" (I would definitely intend to receive it, maybe I would receive it, I would have no intentions of receiving it).

**Vaccine hesitancy (3 items).** We asked 3 questions to measure vaccine hesitancy: 2 questions related to COVID-19 vaccines during pregnancy, and one related to past vaccination behavior, and these were COVID-19 specific as this analysis was conducted within the context of a larger study that aimed to understand COVID-19 vaccine acceptance during pregnancy. We asked women to rate their level of agreement with the following statements on a 4-point Likert scale: "*I do not want to put the COVID-19 vaccine into my body when I am pregnant because I think it is unnatural*" and "*Vaccines improve your body's ability to fight off diseases; this is known as immunity. I believe it is better for my body to develop immunity by getting sick than by getting the COVID-19 vaccine.*" (strongly agree, agree, disagree, strongly disagree). Finally, we asked about past vaccination behavior: "*Have you ever delayed getting a recommended vaccine or decided not to get a recommended vaccine for reasons other than illness or allergy*?" (yes, no, don't know).

**Family influence related to future maternal vaccines (1 item).** We asked women their level of agreement using a 4-point Likert scale with the following statement: "*My family would encourage me to get any vaccine that was recommended during my pregnancy*" (strongly agree, agree, disagree, strongly disagree).

**Priority maternal vaccine and decision-making (2 items).** After providing brief descriptions of the diseases the vaccines could potentially prevent, we asked women which of the four potential future maternal vaccines to select the one that they were most interested in: "*If it were to become available in your country, which one of the vaccines (tuberculosis, malaria, Group B streptococcus, respiratory syncytial virus) would you be most interested in receiving during pregnancy*?" (tuberculosis vaccine, malaria vaccine, Group B streptococcus vaccine (against blood infections and meningitis in babies), respiratory syncytial virus vaccine (against wheezing and pneumonia in children). We asked them to rank in order the reasons that they would want that particular vaccine: "*For the vaccine that you chose in the last question, we have listed multiple reasons why people may want to get that vaccine. Please rank these reasons in order, with 1 being the most important aspect to you for getting the vaccine, to 5 being the least relevant factor in your decision to get the vaccine.*" (I want to protect myself against the disease, I want to protect my baby against the disease, I want to reduce my risk of spreading the disease to others (to protect my family, friends, and others against the disease), my doctor would want me to get the vaccine, my partner and/or my family would want me to get the vaccine).

## Data analysis

After reviewing the data for completeness, we followed a standard protocol for data cleaning, including managing missing data. We calculated frequencies and developed histograms for each variable. Descriptive statistics of sociodemographic variables were calculated for individual countries. Chi-square tests were used for all comparisons between countries. For the variables using 4-point Likert scales, the most extreme responses (strongly agree or strongly disagree) were the least frequent therefore we collapsed these into two categories: agree (strongly agree and agree) and disagree (strongly disagree and disagree). Responses to ranking questions were summarized by the mean rank for each reason selected overall and by country, and the percent of respondents that selected each reason as their most important reason. Data analysis was done in Stata (v18, College Station, TX) and R Statistical Software (v4.4.2, R Core Team 2024).

## Results

There were 1,603 women that participated in our study. Most of the participants were aged 25–34 (55.2%), married or cohabitating (82.6%), had no children under the age of 18 (39.9%), and had completed secondary/high school (47.8%). See Table 1 for sociodemographic characteristics of the study sample.

We asked participants to choose which factors (up to three factors) were most important related to informing their decision-making for vaccines in pregnancy. Overall, the most frequently selected factor was vaccine safety for the baby (65.3%) followed closely by efficacy for the baby (57.6%), while the least selected reason related to family recommendations (21.5%). For the most part, country specific data followed similar trends with a few exceptions (Fig 1). In Kenya, vaccine safety for the mother was chosen most frequently, while in Ghana, vaccine efficacy for the mother was chosen

**Table 1. Sociodemographic characteristics overall and by country.**

| | Overall (N = 1603) | Brazil (N = 402) | Ghana (N = 401) | Kenya (N = 400) | Pakistan (N = 400) |
|---|---|---|---|---|---|
| Age, n(%) | | | | | |
| < 18 and emancipated | 6 (0.4) | 6 (1.5) | 0 (0.0) | 0 (0.0) | 0 (0.0) |
| 18-24 | 402 (25.1) | 109 (27.1) | 80 (20.0) | 61 (15.2) | 152 (38.0) |
| 25-34 | 885 (55.2) | 198 (49.3) | 224 (55.9) | 249 (62.3) | 214 (53.5) |
| 35-49 | 310 (19.3) | 89 (22.1) | 97 (24.2) | 90 (22.5) | 34 (8.5) |
| Marital status, n(%) | | | | | |
| Single (never married) | 257 (16.0) | 125 (31.1) | 91 (22.7) | 41 (10.2) | 0 (0.0) |
| Married or cohabitating | 1324 (82.6) | 263 (65.4) | 303 (75.6) | 358 (89.5) | 400 (100.0) |
| Divorced, separated, or widowed | 22 (1.4) | 14 (3.5) | 7 (1.7) | 1 (0.2) | 0 (0.0) |
| Pregnancy Trimester, n(%) | | | | | |
| First trimester (1–12 weeks) | 446 (27.8) | 133 (33.1) | 49 (12.2) | 131 (32.8) | 133 (33.2) |
| Second trimester (13–26 weeks) | 572 (35.7) | 135 (33.6) | 170 (42.4) | 134 (33.5) | 133 (33.2) |
| Third trimester (from 27 weeks) | 585 (36.5) | 134 (33.3) | 182 (45.4) | 135 (33.8) | 134 (33.5) |
| Living children under 18 years of age, n(%) | | | | | |
| None | 639 (39.9) | 160 (39.8) | 151 (37.7) | 165 (41.2) | 163 (40.8) |
| One | 472 (29.4) | 135 (33.6) | 102 (25.4) | 132 (33.0) | 103 (25.8) |
| Two | 313 (19.5) | 70 (17.4) | 87 (21.7) | 77 (19.2) | 79 (19.8) |
| Three | 118 (7.4) | 21 (5.2) | 41 (10.2) | 20 (5.0) | 36 (9.0) |
| Four or more | 61 (3.8) | 16 (4.0) | 20 (5.0) | 6 (1.5) | 19 (4.8) |
| Education level, n(%) | | | | | |
| No formal schooling | 18 (1.1) | 0 (0.0) | 3 (0.7) | 0 (0.0) | 15 (3.8) |
| Less than primary school | 85 (5.3) | 31 (7.7) | 12 (3.0) | 9 (2.2) | 33 (8.2) |
| Primary school completed | 190 (11.9) | 65 (16.2) | 30 (7.5) | 37 (9.2) | 58 (14.5) |
| JHS/SHS school completed | 767 (47.8) | 257 (63.9) | 245 (61.1) | 83 (20.8) | 182 (45.5) |
| College/University completed | 441 (27.5) | 44 (10.9) | 90 (22.4) | 208 (52.0) | 99 (24.8) |
| Post-graduate degree | 97 (6.1)* | 5 (1.2)* | 16 (4.0) | 63 (15.8) | 13 (3.2) |

*n=5 missing.

most frequently. While a family recommendation was the lowest selected factor overall, Pakistan had a greater number of participants selecting family and doctor recommendation compared to the other countries.

We then examined vaccine intention for each of the four vaccines by country. To illustrate the range of vaccine intentions, Fig 2 shows future maternal vaccine intentions per vaccine by country, denoting the number of participants by country that would intend to definitely receive the vaccine, maybe receive the vaccine, and would have no intentions in receiving the vaccine during pregnancy. There were some interesting country differences: the majority of participants in Pakistan and Brazil indicated that they would definitely intend to receive all four vaccines during pregnancy. In Ghana, there was a higher proportion of participants that indicated they would maybe receive the TB, RSV, and GBS vaccine, compared to other countries. The largest number of participants not intending to receive the TB vaccine was in Kenya. By country, the majority of participants in Brazil definitely intended to receive a GBS vaccine, followed closely by an RSV vaccine; the majority of Ghanaian participants definitely intended to receive a malaria vaccine; in Kenya, the majority of participants definitely intended to receive an RSV vaccine, followed closely by a GBS vaccine; and in Pakistan, the majority of participants definitely intended to receive an RSV vaccine, followed closely by a GBS vaccine. Overall, across all four countries and vaccines, all had the greatest number of participants in the definitely yes and maybe categories as compared to the no category.

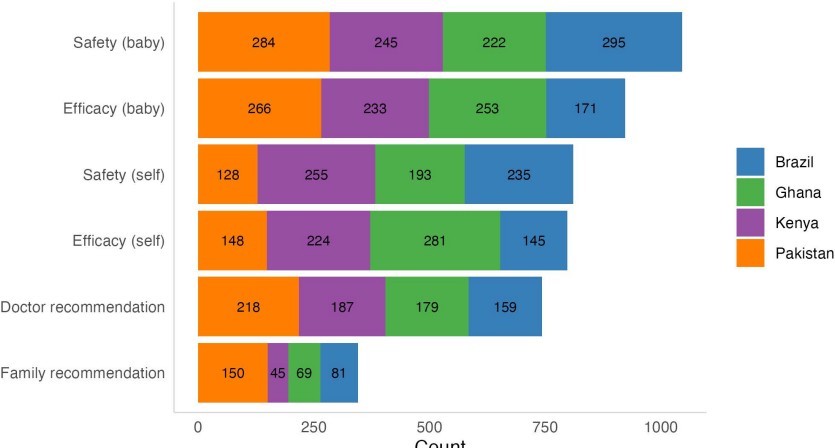

**Fig 1. Priority factors for informing decision-making for future maternal vaccines.**

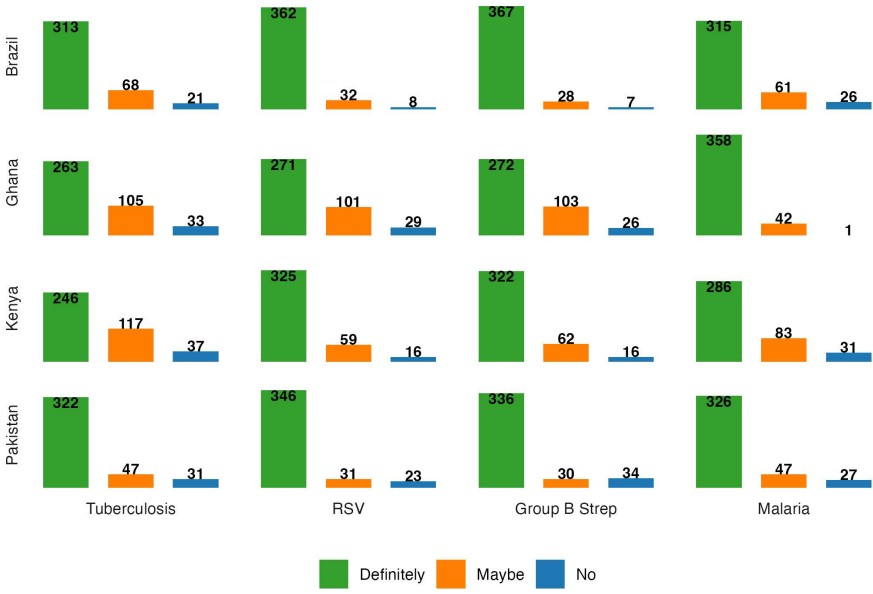

**Fig 2. Intentions to receive vaccines during pregnancy across countries.**

We examined vaccine attitudes broadly among our participants overall and by country (Table 2). We asked three questions related to vaccine hesitancy: 40.1% of our participants agreed that vaccines are unnatural, 37.5% agreed that the body should develop natural immunity, and 19.2% had delayed a recommended vaccine. There were some notable differences by country. Participants in Brazil were less likely to believe that the body should develop natural immunity compared to the other countries (12.9% vs 36.4% in Ghana, 48.8% in Kenya, and 52.0% in Pakistan). Very few participants in Pakistan indicated that they had ever delayed a recommended vaccine (2.0%). Related to family influence, overall, approximately 82% of participants agreed that their family would encourage them to get a vaccine recommended during pregnancy, ranging from 73% in Ghana to 92% in Pakistan. Related to future vaccine intentions, the proportion of

PLOS Global Public Health

**Table 2. Attitudes toward vaccines during pregnancy overall and by country.**

| | Overall (N = 1603) | Brazil (N = 402) | Ghana (N = 401) | Kenya (N = 400) | Pakistan (N = 400) | p-value* |
|---|---|---|---|---|---|---|
| Vaccine hesitancy: agree or yes, n(%) | | | | | | |
| The vaccine is unnatural | 643 (40.1) | 107 (26.6) | 156 (38.9) | 153 (38.2) | 227 (56.8) | <0.001 |
| Immunity is better than a vaccine | 601 (37.5) | 52 (12.9) | 146 (36.4) | 195 (48.8) | 208 (52.0) | <0.001 |
| Delayed vaccination | 307 (19.2) | 135 (33.6) | 65 (16.2) | 99 (24.8) | 8 (2.0) | <0.001 |
| Family influence: agree, n(%) | | | | | | |
| My family would encourage me to get any vaccine | 1315 (82.0) | 339 (84.3) | 291 (72.6) | 319 (79.8) | 366 (91.5) | <0.001 |
| Future maternal vaccines intentions: definitely intend to receive, n(%) | | | | | | |
| Tuberculosis vaccine | 1144 (71.4) | 313 (77.9) | 263 (65.6) | 246 (61.5) | 322 (80.5) | <0.001 |
| Malaria vaccine | 1285 (80.2) | 315 (78.4) | 358 (89.3) | 286 (71.5) | 326 (81.5) | <0.001 |
| GBS vaccine | 1297 (80.9) | 367 (91.3) | 272 (67.8) | 322 (80.5) | 336 (84.0) | <0.001 |
| RSV vaccine | 1304 (81.3) | 362 (90.0) | 271 (67.6) | 325 (81.2) | 346 (86.5) | <0.001 |
| Priority maternal vaccine: top choice, n(%) | | | | | | |
| Tuberculosis vaccine | 298 (18.6) | 50 (12.4) | 58 (14.5) | 73 (18.2) | 117 (29.2) | |
| Malaria vaccine | 413 (25.8) | 15 (3.7) | 222 (55.4) | 104 (26.0) | 72 (18.0) | |
| GBS vaccine | 483 (30.1) | 205 (51.0) | 64 (16.0) | 104 (26.0) | 110 (27.5) | |
| RSV vaccine | 408 (25.5) | 131 (32.6) | 57 (14.2) | 119 (29.8) | 101 (25.2) | <0.001 |

*p-values are from chi-squared tests comparing responses between countries.

GBS: Group B streptococcus, RSV: respiratory syncytial virus.

For all the variables using 4-point Likert scales, response options were collapsed into two categories: agree (strongly agree and agree) and disagree (strongly disagree and disagree).

participants who indicated that they would definitely intend to receive TB, malaria, GBS, or RSV vaccines in pregnancy was generally high (ranging from 71.4% to 81.3% across the various vaccines). There were differences at the country level, with participants in Ghana had the lowest intentions to definitely receive any of the four vaccines except for the malaria vaccine, while participants in Brazil had the lowest intentions to definitely receive the malaria vaccine across the four vaccines in pregnancy.

To understand the salience of decision-making factors related to vaccines in pregnancy, we asked women select the vaccine they were most interested in receiving during pregnancy: TB, malaria, GBS, or RSV. The proportion of overall participants that selected each future vaccine in pregnancy is summarized in Fig 3. Almost one-third of our overall sample (30.1%) prioritized receiving a GBS vaccine, followed by malaria (25.8%), RSV (25.5%), and TB (18.6%) vaccine in pregnancy (Fig 3). At the country-level, the biggest proportion of women in Brazil (51.0%) and Pakistan (27.5%) chose GBS as their priority vaccine. More than half women in Ghana (55.4%) prioritized the malaria vaccine, while in Kenya 29.8% of women selected the RSV vaccine as their priority vaccine to receive in pregnancy.

For the vaccine a person chose as their priority to receive during pregnancy, we asked participants to then rank factors in order of importance for receiving their chosen priority vaccine. We constructed a heatmap to illustrate their rankings (Fig 4). Overall, participants indicated that protecting the baby was most important (50.2%), followed by protecting themselves (36.4%), followed by stopping the spread of disease (5.8%). By country, these were also the first and second factors chosen across Brazil, Kenya, and Pakistan. In Ghana, the most important factor was protecting themselves, followed by protecting the baby, and then stopping the spread of the disease. In Pakistan, the most important factors were protecting the baby, followed by protecting themselves, and in third place a doctor's recommendation.

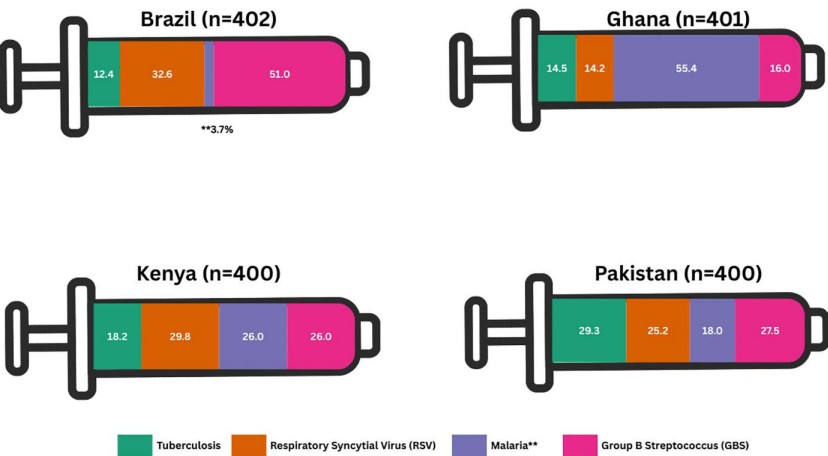

**Fig 3. Priority future vaccine for use in pregnancy.**

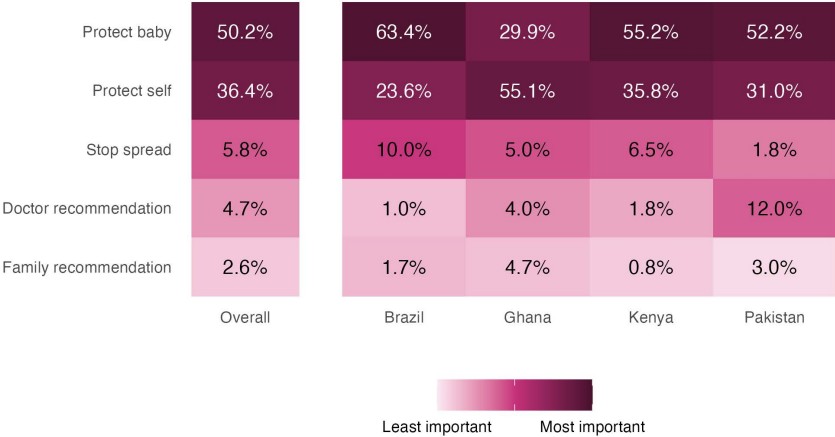

**Fig 4. Most important factors for taking their chosen priority vaccine.**

## Discussion

Our study indicated that vaccine safety for the baby and protecting the baby were the most important factors in vaccine acceptance in pregnancy, followed by self-protection and stopping the spread of disease. Pregnant women are interested in receiving various vaccines during pregnancy, which is important as several new adult vaccines are on the horizon, some of which are designed specifically for pregnancy and others that can be used in pregnancy. Understanding the attitudes of potential vaccine beneficiaries at higher risk for diseases, such as pregnant women, is critical for informing clinical trial design and, in the long term, vaccine acceptance.

Our results align with the broader literature in highlighting high vaccine acceptance during pregnancy, particularly for vaccines developed specifically for use in pregnancy. This includes GBS, where other studies found that acceptance was driven by perceived benefits and low barriers [39]. The high interest in GBS vaccines from participants from Brazil may stem from broader concern about meningitis and sepsis ("blood infections") in newborns; GBS screening and IAP are inconsistent in Brazil, as in many LMICs [40,41], but recent studies have indicated a relatively high prevalence and

severity of GBS in the country and growing concern about antibiotic resistance [23,42–47]. Together, these may have helped drive the relatively higher interest in GBS vaccines in Brazil compared to other vaccines and study countries. With GBS vaccines on the horizon, further research could help prepare Brazilian communities and health workers to leverage this demand and improve GBS prevention.

Similarly, intention to receive the RSV vaccine is consistent with high acceptance rates reported in Canada and the United States, both over 50%, where the perception of RSV as a serious illness strongly influenced vaccination intentions [48–50]. However, the study also revealed notable country-level differences in vaccine priorities, such as Kenyan participants favoring the RSV vaccine and Ghanaian participants prioritizing the malaria vaccine. These variations suggest the importance of context-specific factors; local disease burden and healthcare contexts significantly shape vaccine intentions.

There was equally as strong support for adult vaccines that can be used in pregnancy. Strong interest in TB vaccination in pregnancy was the most common responses across all four countries. This aligns with reports among women in Amhara, Ethiopia, who expressed strong acceptance due to perceived benefits for child protection and societal health [51]. Furthermore, research on TB vaccine acceptance in pregnancy is currently sparse and our findings provide new insights. There was also strong interest in receiving a malaria vaccine in pregnancy, with notable differences in priority by countries with different disease burdens. The research community has agreed that including pregnant women in malaria vaccine efficacy trials should be done, which will provide the needed safety data to help decision making [52]. Malaria vaccines intended specifically for use in pregnancy are also possible, but early in clinical development [53,54].

Our study highlights the important role of vaccine safety and efficacy in shaping pregnant women's decisions regarding immunization. Our finding of the importance of vaccine safety and efficacy for the baby followed by vaccine safety for the mother aligns with prior studies indicating that pregnant women are more likely to accept vaccines when they have confidence in their safety and perceive a benefit to their babies [55–57].These results underscore the need for pregnancy-specific safety data, which can only be generated through the active inclusion of pregnant women in clinical trials. Although trials are enrolling pregnant women, their participation is variable which can be improved by incorporating their concerns and priorities in trial design to generate robust evidence in this population [58–61]. Previous research has similarly found that maternal vaccine acceptance is influenced by vaccine safety, vaccine effectiveness, lack of vaccine knowledge, mistrust of vaccines, perceived disease severity, and lack of recommendation to receive vaccines [62]. This suggests that pregnant women may perceive varying levels of risk and benefit associated with different diseases, further emphasizing the need for targeted educational interventions and communication regarding disease severity and vaccine effectiveness.

Furthermore, our study identified a significant level of vaccine hesitancy, defined as agreeing that vaccines are unnatural, believing that natural immunity is preferable, and ever having previously delayed a recommended vaccine. These attitudes, while consistent with region-specific trends of vaccine hesitancy, were less prevalent in Pakistan, where family encouragement for vaccination was highest. These concerns reinforce the ethical argument that excluding pregnant women from trials results in greater harm, as it deprives them of reliable safety data and contributes to uncertainty [63].

Vaccine uptake is multi-factorial. We examined factors related to intentions to get future recommended vaccines during pregnancy which are important when planning new vaccine introductions [64]. Given that pregnant women report that safety and protection for themselves and their newborns are the primary salient factors in the decision-making process for both existing and new vaccines, future research should focus on developing appropriate and culturally relevant messaging to ensure proper dissemination and uptake. Notable, also, is the role of healthcare providers in immunization among pregnant and post-pregnant women. Further research should focus on understanding their perceptions and motivations in providing recommendations among pregnant and recently pregnant women. Similarly important, is to look at different intersectional factors that might influence maternal immunization. This would include gathering perspectives of women

living in rural areas, women with different abilities, indigenous women, and women of different ethnic and racial backgrounds, as well as any intersectional factors [65]. Overall, our study underscores the importance of contextual factors in shaping vaccine intentions, complementing previous literature while highlighting unique regional variations.

This study has limitations. Participants were recruited in health facilities while seeking antenatal care, which limited our sample selection to pregnant women seeking preventative care. Access and scale of ANC services in each setting likely influenced the study sample. Clinics were chosen to represent different socioeconomic groups of participants but are likely not representative. Administering the surveys in health facilities may have led to social desirability bias. Data are self-reported and are subject to recall bias. This analysis was done as part of a larger study that focused on COVID-19 vaccination for pregnant women, and as such, constructs were focused on attitudes toward COVID-19 vaccines. We did not provide detail on the four future vaccines or diseases to survey respondents, beyond a simple description of RSV and GBS disease; without such background, limited respondent knowledge of these four diseases in pregnancy and neonates may have influenced responses. Despite these limitations, our study has several strengths. This is one of the first studies to examine pregnant women's attitudes toward future vaccines across several different countries, which is critical for new vaccine sensitization, delivery, and uptake.

Participants' willingness to receive a variety of vaccines while pregnant is promising and indicates the potential for high uptake. Several new adult vaccines are on the horizon, some targeted specifically for use in pregnancy. Understanding the attitudes of potential vaccine beneficiaries at higher risk for diseases that vaccines may be able to prevent can help inform clinical trial study design, and in the long-term, demand generation strategies for successful uptake.

## Author contributions

**Conceptualization:** Rupali Limaye, Jessica Schue, Vanessa Brizuela, Ruth Karron.

**Data curation:** Rupali Limaye, Jessica Schue.

**Formal analysis:** Rupali Limaye, Jessica Schue.

**Funding acquisition:** Rupali Limaye, Ruth Karron.

**Investigation:** Rupali Limaye.

**Methodology:** Rupali Limaye, Jessica Schue, Ruth Karron.

**Project administration:** Rupali Limaye, Jessica Schue, Ruth Karron.

**Resources:** Rupali Limaye, Ruth Karron.

**Software:** Rupali Limaye.

**Supervision:** Rupali Limaye, Jessica Schue, Ruth Karron.

**Validation:** Rupali Limaye, Jessica Schue, Berhaun Fesshaye.

**Visualization:** Rupali Limaye, Jessica Schue, Berhaun Fesshaye, Prachi Singh, Emily Miller.

**Writing – original draft:** Rupali Limaye, Jessica Schue, Berhaun Fesshaye, Prachi Singh, Emily Miller, Renato Souza, Saleem Jessani, Marleen Temmerman, Caroline Dinam Badzi, Molly Sauer, Vanessa Brizuela, Ruth Karron.

**Writing – review & editing:** Rupali Limaye, Jessica Schue, Berhaun Fesshaye, Prachi Singh, Emily Miller, Renato Souza, Saleem Jessani, Marleen Temmerman, Caroline Dinam Badzi, Molly Sauer, Vanessa Brizuela, Ruth Karron.

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
