## [Decision Letter · Decision Letter 0]

14 Jul 2025

PGPH-D-25-00806

Pregnant women’s attitudes and intentions toward tuberculosis, malaria, group B streptococcus, and respiratory syncytial virus vaccines in pregnant: Findings from pregnant women living in Brazil, Ghana, Kenya, and Pakistan

Dear Dr. Limaye,

Thank you for submitting your manuscript to PLOS Global Public Health. After careful consideration, we feel that it has merit but does not fully meet PLOS Global Public Health’s publication criteria as it currently stands. Therefore, we invite you to submit a revised version of the manuscript that addresses the points raised during the review process.

The reviewers raised concerns about methodological clarity, the rationale behind the survey, and the analysis. They requested clarification of the inclusion criteria (e.g., prior awareness of COVID-19 vaccines), justification for vaccine choices, and an explanation for dichotomizing Likert responses. Additionally, they questioned cross-country comparisons in a primarily attitude-descriptive survey. Ethical issues regarding participant expectations and the sharing of missing data information also need to be addressed.

We look forward to receiving your revised manuscript.

Kind regards,

Giridhara Rathnaiah Babu, MBBS, MPH, PhD

Academic Editor

Journal Requirements:

2. Please provide separate figure files in .tif or .eps format.

3. In the online submission form, you indicated that "Data will be made available upon reasonable request."

a. In a public repository,

b. Within the manuscript itself, or

c. Uploaded as supplementary information.

4. Please provide a/amend your detailed Financial Disclosure statement. This is published with the article. It must therefore be completed in full sentences and contain the exact wording you wish to be published.

**Please only choose the relevant sentences from below**

a. Please clarify all sources of funding (financial or material support) for your study. List the grants (with grant number) or organizations (with url) that supported your study, including funding received from your institution.

b. State the initials, alongside each funding source, of each author to receive each grant.

c. State what role the funders took in the study. If the funders had no role in your study, please state: “The funders had no role in study design, data collection and analysis, decision to publish, or preparation of the manuscript.”

d. If any authors received a salary from any of your funders, please state which authors and which funders.

Additional Editor Comments (if provided):

The reviewers raised concerns about methodological clarity, the rationale behind the survey, and the analysis. They requested clarification of the inclusion criteria (e.g., prior awareness of COVID-19 vaccines), justification for vaccine choices, and an explanation for dichotomizing Likert responses. Additionally, they questioned cross-country comparisons in a primarily attitude-descriptive survey. Ethical issues regarding participant expectations and the sharing of missing data information also need to be addressed.

Reviewers' comments:

Reviewer's Responses to Questions

**Comments to the Author**

1. Does this manuscript meet PLOS Global Public Health’s publication criteria?

Reviewer #1: Partly

Reviewer #2: Yes

Reviewer #3: Yes

2. Has the statistical analysis been performed appropriately and rigorously?

Reviewer #1: No

Reviewer #2: Yes

Reviewer #3: I don't know

3. Have the authors made all data underlying the findings in their manuscript fully available (please refer to the Data Availability Statement at the start of the manuscript PDF file)?

Reviewer #1: No

Reviewer #2: Yes

Reviewer #3: Yes

4. Is the manuscript presented in an intelligible fashion and written in standard English?

Reviewer #1: Yes

Reviewer #2: Yes

Reviewer #3: Yes

Reviewer #1: Thank you for asking me to review this paper which reports the findings of a cross-sectional survey, completed by pregnant women in four countries on their attitudes towards vaccinations. The paper reports an important area of study and I applaud the authors for their important message about the importance of inclusion of pregnant women in clinical trials. However, there are several issues I have identified in the paper, mostly around methods and expansion on why some of the questions have been asked, which would require revision prior to publication.

Title

• Title should include that this is a cross-sectional survey, i.e. “Findings from a cross-sectional survey of pregnant women…..”

Background

• Worth mentioning in the background that GBS can also result in death of the baby

• Background sets the scene nicely but would be helpful to expand a little on the final paragraph re women’s attitudes to vaccines – briefly mention any previous work and why this is an important question to answer, i.e. even if vaccines are made available in LMICs important to know whether they would be considered acceptable to those who they are intended for

• As this paper reports the results of a cross-sectional survey, as part of a wider programme of work, this should be specified at the end of the background section, rather than in the ‘participants, study setting and recruitment’ sub-section of Methods. It will make it clear that this study reports only the survey data, and that data collected from qualitative interviews is reported separately.

• The background section should include a clearly framed research question, with clearly specified objectives for the study.

Methods

• Please add a sentence that summarises the total number of sites/facilities that were involved

• Page 7, the sentence regarding sample size is unclear – ‘proportion of respondents with an attitude’. Surely everyone had an attitude, positive or negative, towards vaccines. Please be clearer here. What were the sample size assumptions based on?

• It’s unclear to me why formal statistical testing was undertaken and looking for a statistical difference between countries. For this kind of study, I would have expected to see descriptive statistics only reported.

• Data collection – were women provided with written participant information? Please make this clear

• Please explain the rationale for having “heard of Covid-19 vaccine” as an eligibility criteria for inclusion

• The countries who provided food box/transport renumeration – was this before or after completion of the survey?

• Future maternal vaccine intentions: Did you collect data on the reasons for the response choice for the question relating to their intentions? i.e. “I would not have no intentions of receiving the vaccine”…. Because……..

• Vaccine hesitancy – please explain the rationale for including a very specific question relating to the Covid-19 vaccination. I appreciate this is likely to set the scene since many women will have heard of this particular vaccine, but I think some context is required. Arguably, getting a vaccine for Covid-19 is quite different to GBS, for example.

• Priority maternal vaccine – please explain the rationale for inclusion of this question in the survey. Can the authors comment on the potential ethical implications of raising hope in women that (some of) these vaccines may become available in the future – how was this mitigated? In addition, did the authors collect any data on the past medical or family medical history? The answer to the question re priority maternal vaccine could be influenced by their own health or that of a family member – i.e. family member may have respiratory problems, so woman may be more likely to choose this vaccine if they thought it may prevent them developing respiratory issues

Results

• P values have been included to show differences between countries - but it's not clear why this is important. The p values don't add anything - it's clear there are differences by looking at the frequencies etc. Seeing whether there is a statistically significant difference between countries is of less importance. I would have expected to see the data in a survey such as this one being presented descriptively (e.g. means, medians etc.). The aims and objectives are not clearly specified - was it always the intention to look for between country differences?

• Table 2 is very unclear

• It would be helpful to add percentages to figure 1

• The quality of figure 3 makes it difficult to read – but I question what this figure adds. It looks like it reports the same data as in figure 2, in an alternative format

• The results present information that should be reported in the methods section. For example, line 381 on page 17. This is one example, but this happens throughout.

Discussion

• Given the apparent emphasis on comparison between countries, I would have expected to see discussion about what this could mean in practice in terms of distribution of vaccinations and the work needed with in-country policy-makers regarding procurement of vaccinations and implementation into the health systems

• Great to see the authors advocating for the inclusion of pregnant women in clinical trials

Other

• It is a journal requirement to make data publicly available – the manuscript does not include details of where the data is shared. In the submission information it states ‘data is available upon reasonable request’

Reviewer #2: Line 44 to 46: vaccine safety for baby has been repeated,

Line 157: you have mentioned COVID 19 vaccine decision making

Line 188: why having heard of the COVID-19 vaccine is an inclusion criteria? Isn't the study about TB, Malaria, GBS and RSV?

Line 259: same as above, the vaccine hesitancy section is all about COVID-19

Line 373: While the study provides useful insights into vaccine prioritization among pregnant women across countries, it remains unclear whether participants had adequate knowledge of the diseases or vaccine safety profiles. The absence of questions exploring their rationale limits the interpretability of these preferences. Most women have received high school education, when presented with a list of 4 vaccines , it is overwhelming and their choices reflect guesswork or based on whatever little they know or don't know about that disease. Without adequate knowledge about the disease, it's burden and consequences being informed, it is difficult to understand the rationale to understand vaccine acceptance study 4 different diseases.

Reviewer #3: In this study Rupali et al have explored the attitudes of pregnant women in 4 countries regarding the uptake of 4 plausible future vaccines. The topic is an important one and the study has been conducted well. Please find my comments and queries below.

Major comments

1. The study has been described as a mixed methods study. However, only quantitative surveys seem to have been conducted. Were there no interviews or focus group discussions done to understand the major emerging themes regarding the attitudes of the participants?

2. Why was knowledge on COVID-19 vaccine a requirement for inclusion in the study?

3. Of the many potential vaccine- preventable diseases, why these specific four vaccines were selected for the purpose of the study?

4. Why was the data collected using Likert scale dichotomized during analysis? This could have led to potential loss of information.

5. It is not clear from the methods section whether any information on the potential effects of these 4 diseases on pregnancy was imparted to the participants. The attitude of the study participants will definitely depend upon their perception of threat from each of these diseases for themselves as well as the baby. It is difficult to imagine that all the study participants had baseline knowledge about diseases caused by GBS or RSV. Even in the case of more well-known diseases like malaria or TB, their effect on pregnancy is not something which is widely known. Please explain.

6. In Discussion section (lines 404 to 408) the authors are attributing the favorable attitude of pregnant women for GBS vaccine to the high prevalence of this infection, antibiotic resistance and the disease related morbidity and mortality. Again, unless the information regarding these was provided to the participants, it is difficult to assume that these factors would have influenced their choices.

Minor comments

1. Please correct the repetition in the following statement found in abstract “Participants indicated that vaccine safety for the baby was the most important factor in their decision-making related to vaccine acceptance, followed by vaccine efficacy for the baby, and then vaccine safety for the baby.”

2. Please rephrase and correct the sentence “The sample size was determined with 80% power to find differences in proportions of respondents with an attitude between two groups with a 5% margin of error.” Proportion with an attitude??

3. Line number 200 is confusing. “members of the team from (blinded for review) through…”

4. “In Kenya, vaccine safety for the mother was chosen most frequently, and in Ghana, vaccine efficacy for the mother was chosen most frequently.” Use of the conjunction AND seems inappropriate here.

5. In lines 360 to 362 the use of terms like “least likely to definitely intend to receive” is very confusing to read.

**Do you want your identity to be public for this peer review?** For information about this choice, including consent withdrawal, please see our Privacy Policy

Reviewer #1: No

Reviewer #2: **Yes:** Deepa R

Reviewer #3: **Yes:** Deepanjali S

---

## [Decision Letter · Decision Letter 1]

6 Oct 2025

Pregnant women’s attitudes and intentions toward tuberculosis, malaria, group B streptococcus, and respiratory syncytial virus vaccines in pregnant: Findings from pregnant women living in Brazil, Ghana, Kenya, and Pakistan

PGPH-D-25-00806R1

Dear Dr. Limaye,

We are pleased to inform you that your manuscript 'Pregnant women’s attitudes and intentions toward tuberculosis, malaria, group B streptococcus, and respiratory syncytial virus vaccines in pregnant: Findings from pregnant women living in Brazil, Ghana, Kenya, and Pakistan' has been provisionally accepted for publication in PLOS Global Public Health.

Best regards,

Giridhara Rathnaiah Babu, MBBS, MPH, PhD

Academic Editor

Reviewer Comments (if any, and for reference):

Reviewer's Responses to Questions

**Comments to the Author**

Reviewer #1: All comments have been addressed

Reviewer #3: All comments have been addressed

publication criteria?

Reviewer #1: Yes

Reviewer #3: Yes

3. Has the statistical analysis been performed appropriately and rigorously?

Reviewer #1: I don't know

Reviewer #3: I don't know

4. Have the authors made all data underlying the findings in their manuscript fully available (please refer to the Data Availability Statement at the start of the manuscript PDF file)?

Reviewer #1: No

Reviewer #3: Yes

5. Is the manuscript presented in an intelligible fashion and written in standard English?

Reviewer #1: Yes

Reviewer #3: Yes

Reviewer #1: Thank you for addressing feedback. I have no further comments.

Reviewer #3: The authors have addressed the issues raised

**Do you want your identity to be public for this peer review?** For information about this choice, including consent withdrawal, please see our Privacy Policy

Reviewer #1: **Yes:** Eleanor Mitchell

Reviewer #3: **Yes:** Deepanjali S
